# Antifungal Activities of *Bacillus mojavensis* BQ-33 towards the Kiwifruit Black Spot Disease Caused by the Fungal Pathogen *Didymella glomerata*

**DOI:** 10.3390/microorganisms10102085

**Published:** 2022-10-21

**Authors:** Bingce Wang, Xia Lei, Jia Chen, Wenzhi Li, Youhua Long, Weizhen Wang

**Affiliations:** 1Research Center for Engineering Technology of Kiwifruit, College of Agriculture, Guizhou University, Guiyang 550025, China; 2Institute of Crop Protection, College of Agriculture, Guizhou University, Guiyang 550025, China

**Keywords:** kiwifruit black spot, *Didymella glomerata*, biocontrol, antifungal activity

## Abstract

‘Hongyang’ kiwifruit (*Actinidia chinensis*, cultivar ‘Hongyang’) black spot disease is caused by the fungal pathogen *Didymella glomerata*, and is a serious disease, causing considerable losses to the kiwifruit industry during growth of the fruit. Hence, we aimed to identify a potential biocontrol agent against *D. glomerata*. In this study, bacterial isolates from the rhizosphere soil of kiwifruit were tested for their potential antifungal activity against selected fungal pathogens. Based on a phylogenetic tree constructed using sequences of 16S rDNA and the *gyrA* gene, BQ-33 with the best antifungal activity was identified as *Bacillus mojavensis*. We evaluated the antagonistic activity and inhibitory mechanism of BQ-33 against *D. glomerata*. Confrontation experiments showed that both BQ-33 suspension and the sterile supernatant (SS) produced by BQ-33 possessed excellent broad-spectrum antifungal activity. Furthermore, the SS damaged the cell membrane and cell wall of the mycelia, resulting in the leakage of a large quantity of small ions (Na^+^, K^+^), soluble proteins and nucleic acids. Chitinase and β-1,3-glucanase activities in SS increased in correlation with incubation time and remained at a high level for several days. An in vivo control efficacy assay indicated that 400 mL L^−1^ of SS completely inhibited kiwifruit black spot disease caused by *D. glomerata*. Therefore, BQ-33 is a potential biocontrol agent against kiwifruit black spot and plant diseases caused by other fungal pathogens. To our knowledge, this is the first report of the use of a rhizosphere microorganism as a biocontrol agent against kiwifruit black spot disease caused by *D. glomerata*.

## 1. Introduction

Kiwifruit (*Actinidia* spp.), such as Chinese kiwifruit, is known as the ‘king of fruits’ due to its richness in vitamin C, potassium, calcium, and carotene [1,2,3]. China has the largest area of kiwifruit plantations, accounting for 72% of the total area of world kiwifruit plantations [4]. With the increase in plantation scale, kiwifruits have become easily affected by various pathogens during the production process. Black spot disease is a common threat to kiwifruits during growth, affecting their appearance and quality and seriously reducing the profitability of kiwifruits in the market. Black spot disease was initially thought to be specific to ‘Cuixiang’ kiwifruit (*Actinidia deliciosa*, cultivar ‘Cuixiang’), but gradually, other kiwifruit varieties were also found to be vulnerable. Kiwifruit black spot can be caused by different fungal pathogens, including *Cladosporium cladosporioides*, *Diaporthe phaseolorum*, and *Trichothecium roseum* [5]. In recent years, *Didymella glomerata* has also been identified as a causal agent of kiwifruit black spot [6], in addition to infecting other plants, such as grape, maize, pistachio, *Cornus officinalis*, and *Sophora tonkinensis* [7,8,9,10,11].

*D. glomerata* has been identified as the main pathogen causing black spot on ‘Hongyang’ kiwifruit. It can infect kiwifruits at different stages of fruit growth [6]. Once infected, the fruit develops slightly sunken black spots on the skin, and the pulp at the lesion area becomes solid and black. In production, farmers mainly rely on synthetic fungicides to reduce the occurrence of kiwifruit black spots and improve fruit quality [12]. Although some chemical agents can quickly and effectively reduce the severity of this disease, the extensive and excessive use of fungicides can accelerate the development of fungicide resistance in fungal pathogens [13]. Additionally, extensive use of such agents may cause food safety concerns and damage to the environment. To address these issues, it is important to develop safe, effective, and environment-friendly biocontrol agents to manage kiwifruit black spot caused by *D. glomerata* [14,15,16].

In agricultural production, biological control has been proposed as a reasonably safe way to limit the effect of plant diseases and has great potential for the sustainability of disease management [17]. Microbial biocontrol agents protect against pathogens through their metabolites, competition for nutrients and space, and parasitism, thus, preventing and controlling diseases. *Bacillus* spp. has attracted wide attention in the field of biological control because of its ability to produce many broad-spectrum antifungal compounds, hydrolases and toxins [18,19]. In recent decades, various biocontrol bacteria have been used to control a variety of plant diseases, including *Bacillus velezensis* and *Bacillus safensis* against foot rot disease in sweet potato, *Bacillus subtilis* to control wheat spot blotch, and *Bacillus cereus* to elicit tomato plant protection against *Fusarium oxysporum* [20,21,22].

Despite the wide use of biocontrols in agricultural production, no studies have reported the biological control of kiwifruit black spot caused by *D. glomerata*. In this study, the biocontrol bacterium *Bacillus mojavensis* strain BQ-33 showed strong activity against *D. glomerata*. The SS of BQ-33 was collected to evaluate its inhibitory activity and mechanism on *D. glomerata*. In addition, an infection assay was conducted to evaluate the biocontrol potential of *B. mojavensis* BQ-33. Overall, this study provides a theoretical basis for the biological control of *D. glomerata* using *B. mojavensis* BQ-33.

## 2. Materials and Methods

### 2.1. Fungal Pathogen

*D. glomerata*, the fungal pathogen of kiwifruit black spot disease, was provided by Professor Caihong Zhong’s team of the Key Laboratory of Plant Germplasm Enhancement and Specialty Agriculture, Wuhan Botanical Garden, Chinese Academy of Sciences [6]. It was routinely cultured on potato dextrose agar (PDA) medium at 28 °C.

### 2.2. Rhizobacteria Isolation and Antifungal Activity Evaluation

Twenty-two rhizosphere soil samples of healthy kiwifruit from Liupanshui, Guizhou, China, were collected for isolation of biocontrol bacteria [23]. From each sample, 20 g of dried and ground soil was mixed with 200 mL of sterile water and incubated in a shaker at 37 °C and 150 rpm for 2 h. The mixtures were subsequently placed in a water bath set at 85 °C for 30 min, followed by homogenizing the solutions by shaking. After the soil had precipitated, 100 μL of the supernatant diluted to 10^−3^, 10^−4^, and 10^−5^ was spread on nutrient agar (NA) medium and incubated at 30 °C for 2 d, using three replicates of each concentration. Single-colony purification was conducted, and the purified colonies were propagated in nutrient broth (NB) medium. Glycerol (25%) was added to the culture systems, and the strains were stored at −80 °C.

The confrontation method was used to test the antagonistic activity of the isolated rhizobacteria [24]. Mycelial plugs (6 mm in diameter) of *Didymella glomerata* were placed at the centre of PDA medium plates. Antagonistic bacterial suspensions, incubated overnight, were applied to sterilised filter paper disks (6 mm in diameter), and four such filter paper disks were placed around the *D. glomerata* mycelial plug at a distance of 25 mm. Petri dishes inoculated only with mycelial plugs were used as controls. The Petri dishes were then placed in an incubator at 28 °C for 4 days. All treatments and controls were analysed in three replicates. The inhibition ratio was calculated by comparing the diameters of colonies in the treatment groups with that in the control groups [25].

### 2.3. Identification of the BQ-33 Strain

The best candidate was further identified according to 16S rDNA and *gyrA* sequence. For molecular identification, the genomic DNA of the antagonistic bacteria was extracted using the FastPure Bacteria DNA Isolation Kit (Vazyme Biotech Co. Ltd., Nanjing, China), according to the manufacturer’s instructions. The 16S rDNA and *gyrA* gene were amplified using primer sets 27F (5′-AGAGTTTGATCCTGGCTCAG-3′) and 1492R (5′-GGTTACCTTGTTACGACTT-3′), and *gyrA*-F (5′-CAGTCAGGAAATGCGTACGTCCTT3′) and *gyrA*-R (5′-CAAGGTAATGCTCCAGGCATTGCT-3′) [26,27]. The PCR products were sequenced by SANGON Biotech Co., Ltd. (Shanghai, China). The obtained sequences were compared with the nucleotide database of the national center for biotechnology information (NCBI), using the Basic Local Alignment Search Tool (BLAST, https://blast.ncbi.nlm.nih.gov/Blast.cgi accessed on 21 July 2022). Reference strain sequences were downloaded from the NCBI database, and a polygene phylogenetic tree was constructed using the maximum likelihood (ML) method in MEGA 7.0 software with bootstrap values calculated, based on 1000 replications [28].

### 2.4. Preparation of QB-33 the Sterile Supernatant (SS)

BQ-33 was inoculated into NB and incubated in a shaker incubator (34 °C, 180 rpm) for 48 h, and, when the OD_600_ reached 0.8, it was added to 600 mL of fresh NB (pH 7.2) at an inoculum volume of 1:100 (*v:v*). Then, the solution was incubated in a shaker incubator at 34 °C and 180 rpm for 4 d. The bacterial suspension was, subsequently, centrifuged at 4 °C and 12,000× *g* for 10 min, and the supernatant was filtered twice through a 0.22 μm sterile syringe filter [29], and the filtered SS was stored at 4 °C.

### 2.5. Test of Antifungal Activity of BQ-33 and Its SS

Some pathogens on kiwifruit can also infect other crops and cause diseases. In order to clarify the spectrum of biological control of BQ-33 in agricultural production, the inhibitory activity of BQ-33 suspension against these fungal pathogens (Table 1) was tested with the method described in Section 2.2. The inhibitory effect of SS was evaluated according to a previously described method [30]. The SS was mixed with the PDA medium in a ratio of 1:5 (*v:v*). Mycelial plugs of the five fungal pathogens were inoculated in the centre of the PDA medium containing SS with three replicates for each species, which were incubated at 28 °C for 4 days. Cells cultured on the PDA medium containing the same amount of NB were used as controls. The inhibition ratio was calculated as described in Section 2.2.

### 2.6. Chitinase and β-1,3-Glucanase Activity Assays

Chitinase activity was determined according to method described in a previous study [35]. Briefly, 50 μL of the supernatant was mixed with 500 μL of colloidal chitin (0.5%), and 450 μL of sodium acetate buffer (0.05 M, pH 5.0) was added to the mixture. The mixture was incubated at 37 °C for 1 h, and 200 μL of NaOH (1 M) was then added to terminate the reaction. Subsequently, the reaction mixture was centrifuged at 4 °C and 10,000× *g* for 5 min. Next, 750 μL of the supernatant was collected and supplemented with 1 mL of Schales’ reagent (0.5 M sodium carbonate and 0.5 g L^−1^ potassium ferricyanide in water). The mixture was then diluted with the addition of 250 μL of deionized distilled water and incubated in boiling water for 15 min. Finally, the OD_420_ value of the treatment solution was determined in three replicates using a microplate reader. The unit of chitinase activity is the amount of enzyme that decomposes chitinase to produce 1 μmol of N-acetylglucosamine per hour per 1 mL of the supernatant at 37 °C, which was expressed as U mL^−1^.

A previously described method was modified to determine the β-1,3-glucanase activity of the BQ-33 supernatant [36]. A total of 50 μL laminarin (10 mg mL^−1^), and 400 μL sodium acetate buffer (0.05 M, pH 5.0), were added to 50 μL of the supernatant. The solution was incubated at 37 °C for 1 h, and then supplemented with 1.5 mL of 3,5-dinitrosalicylic acid (DNS) to terminate the reaction. The OD_550_ value of the treatment solution was determined in three replicates using a microplate. The unit of β-1,3-glucanase activity was defined as the amount of enzyme that decomposed β-1,3-glucanase to produce 1 μmol of glucose per hour per 1 mL of the supernatant at 37 °C, which was expressed as U mL^−1^.

### 2.7. Changes in Cell Permeability

Three 6-mm mycelial disks of *D. glomerata* were transferred to 100 mL potato dextrose broth (PDB) and placed on a constant temperature shaker (150 rpm at 28 °C). After incubation for 3 days, three layers of sterile gauze were used to filter and collect the mycelia, and the collected mycelia were washed three times with sterile water. Then, 1 g of fresh mycelia were suspended in 50 mL PDB medium containing 100 mL L^−1^, 200 mL L^−1^, and 400 mL L^−1^ SS, which were incubated in a shaker at 28 °C and 150 rpm. In the control, mycelia were incubated in PDB without SS. The supernatant was collected after 0, 12, 24, 36, 48, 60 and 72 h of incubation, the relative conductivity was calculated according to the method of Mo et al. [37]. The soluble protein was detected with the Soluble Protein Extraction Kit (SanGon Biotech, Shanghai, China). The degree of nucleic acid leakage was detected at the wavelength of 260 nm absorbance [38]. There were three replicates for each test.

### 2.8. Effects of SS on Cell Structures of Fungal Pathogens

#### 2.8.1. Effects on Cell Walls

Calcofluor white (CFW) was used to detect changes in the integrity of cell walls of *D. glomerata* [39]. Three mycelial plugs of *D. glomerata* were inoculated into 150 mL of PDB medium. After 3 days of incubation in a shaker at 28 °C and 150 rpm, the mycelia were collected by filtering through gauze, and a portion of the mycelia was placed into 50 mL of PDB medium containing 200 mL L^−1^ of SS, using three replicates, and incubated in a shaker at 28 °C and 150 rpm for 12 h. Mycelia incubated in PDB without SS were used as controls. A small quantity of mycelia was stained with 5 μL of CFW in the dark for 10 min and then washed three times with a washing solution. The integrity of the cell wall was observed and photographed using a confocal laser scanning microscope (NE 910-FL, Ningbo Yongxin Optics Co., Ltd., Ningbo, China) at 355–400 nm.

#### 2.8.2. Effects on Cell Membranes

Propidium iodide (PI) was used to detect the integrity of the cell membrane [40]. A part of the mycelia was placed in 50 mL of PDB medium containing 100 mL L^−1^, 200 mL L^−1^, and 400 mL L^−1^ SS, using three replicates, which were incubated in a shaker at 28 °C and 150 rpm for 12 h. Again, mycelia incubated in PDB without SS were used as a control group. The mycelia were collected by centrifugation at 4 °C and 10,000× *g* for 5 min. First, the treated mycelia were washed three times with PBS (0.02 M, pH 7.0). After washing, 5 μL PI was added and the solution was left to stain in the dark for 20 min, after which the mycelia were washed again with PBS (0.02 M, pH 7.0). CLSM was used to observe the fluorescence effect at wavelengths of 536–617 nm to determine the integrity of the cell membrane.

### 2.9. In Vivo Control Effect of SS

Healthy kiwifruit (90 days after fruit setting), with consistent maturity, were collected. The surfaces of the healthy kiwifruit were cleaned with sterile water, soaked in 75% ethanol for 1 min, and then cleaned again with sterile water three times. A syringe needle was used to create four puncture wounds with a depth of 2 mm within a circular area (diameter 5 mm). The mycelial plugs of *D. glomerata* were inoculated onto the stabbed circular area of kiwifruit, which were placed in a sterile fresh-keeping box. Each box was filled with six kiwifruits, and these kiwifruits were then sprayed with SS solutions at concentrations of 400 mL L^−1^, 200 mL L^−1^, and 100 mL L^−1^. Each treatment was repeated three times. In the control group, spraying was performed with sterile water. The fresh-keeping boxes were placed in an artificial climate chamber adjusted to maintain a temperature of 28 °C, a humidity of 75%, and a 14 h/10 h light–dark photoperiod. Disease progression was observed after six days, and the sizes of the lesions were measured to calculate the control efficacy [41].

### 2.10. Statistical Analyses

All the collected data were analysed with ANOVA. All the figures were drawn using Origin 2021 software. Data obtained from the replicates of each experiment are represented in the graphs as the mean ± standard error (SE).

## 3. Results

### 3.1. Screening of Biocontrol Bacteria

A total of 78 bacterial strains were isolated from 22 kiwifruit rhizosphere soil samples, of which 36 displayed inhibition ratios > 50% against *D. glomerata* (Appendix A). These included 11 strains with inhibition ratios of 50–60%, 16 strains with inhibition ratios of 60–70%, and 9 strains with inhibition ratios > 70%, among which *B. mojavensis* BQ-33 showed the highest antifungal activity against *D. glomerata* (81.62%). Therefore, BQ-33 was selected for further analysis and identification. The optimal growth of BQ-33 was achieved at a temperature of 34 °C, using an NB fermentation medium and a fermentation pH of 7.2.

### 3.2. Identification of Strain BQ-33

To identify the bacterial strain BQ-33 at the species level, the 16S rDNA and *gyrA* gene sequences were amplified using PCR and Sanger sequencing produced two nucleotide sequences with lengths of approximately 1435 bp and 985 bp. The sequences of 16S rDNA (ON231801) and *gyrA* (ON245037) were uploaded to the GenBank database. BLAST analysis showed that the 16S rDNA and *gyrA* sequences of BQ-33 shared a high degree of homology with *B. mojavensis* UCMB5075 (accession number: CP051464), with 98.96% similarity between 16S rDNA sequences and 98.69% similarity between *gyrA* sequences. A phylogenetic tree was constructed using 16S rDNA and *gyrA* gene sequences of BQ-33, along with a few reference isolates obtained from GenBank (Appendix A). Phylogenetic analysis further confirmed that BQ-33 clustered monophyletically with a strain of *B. mojavensis* UCMB5075 (Figure 1). Based on these results, the isolated strain BQ-33 was identified as *B. mojavensis*.

### 3.3. Antifungal Spectrum of BQ-33 Regarding Kiwifruit Diseases

Evaluation of the inhibitory effect towards fungal pathogens showed that both BQ-33 suspension and SS displayed excellent bioactivity against the main fungal pathogens of the fruits, leaves, and branches of kiwifruit plants. The growth of the five pathogens was significantly inhibited by BQ-33 suspension, resulting in significantly reduced mycelial growth (Figure 2A) and inhibition ratios of 70.11–81.26% (Table 2). Compared with the control, 200 mL L^−1^ SS also had a good inhibitory effect on the growth of fungal pathogens (Figure 2B). The inhibitory effect of SS on *D. glomerata* was the highest, with an inhibition ratio of 83.43% (Table 2).

### 3.4. Enzymatic Activities of BQ-33 SS

The chitinase and β-1,3-glucanase activities of BQ-33 SS were monitored for 12 d. The hydrolase activities of SS rapidly increased during the first two days and stayed relatively consistent over the next 10 days. The activities of chitinase and β-1,3-glucanase increased significantly from 0 to 3 days (Figure 3). From day 3 to 8, chitinase activity increase was generally slow, peaking on day 8 (Figure 3A). The β-1,3-glucanase activity maintained an increasing trend until day 4 and reached a plateau on days 6–8 (Figure 3B).

### 3.5. Effect of SS on the Permeability of D. glomerata Cells

In order to clarify the inhibition mechanism of BQ-33 SS on *D. glomerata*, the changes of relative conductivity, nucleic acid and soluble protein content in *D. glomerata* culture medium after SS treatment were determined. The results indicated that SS affected the permeability of *D. glomerata* cells. With the increase of SS concentration, the relative conductivity gradually increased, and the leakage of nucleic acid and soluble protein also increased in a similar pattern (Figure 4), which caused the *D. glomerata* mycelia to be unable to grow normally.

### 3.6. BQ-33 SS Influence on Fungal Cells

Fluorescence of the cell wall was observed after treatment with CFW. The control group displayed strong fluorescence, which was consistent with the fact that the mycelial cell wall was intact. However, the fluorescence of a part of the mycelia in the treatment group was weak or completely missing (Figure 5), indicating that the cell wall of *D. glomerata* mycelia was seriously damaged by the treatment with 200 mL L^−1^ of SS. PI could enter the cell through the damaged cell membranes and combine with nucleic acids to produce red fluorescence. The mycelia in the control group showed almost no fluorescence upon PI staining (Figure 6). Under treatments with three different concentrations of SS, the mycelia of *D. glomerata* showed red fluorescence with different intensities and the fluorescence intensity of mycelia increased in correlation with increasing SS concentration.

### 3.7. Biocontrol Efficacy of BQ-33 SS towards Kiwifruit Black Spot Disease

To evaluate the biocontrol efficacy of BQ-33 SS towards black spot disease of kiwifruit fruits, different concentrations of SS were tested on kiwifruit inoculated with *D. glomerata*. After five days of incubation under the same storage condition, large black spots appeared on the skin of the water-treated kiwifruits (Figure 7). The symptoms of kiwifruit black spots were relieved by SS treatments, and the fruits treated with 400 mL L^−1^ of SS did not develop any symptoms, displaying 100% control efficacy (Table 3). After treatments with 200 mL L^−1^ and 100 mL L^−1^ of SS, black spots of varying severities appeared on the fruits, and the control efficacies were 74.75% and 47.52%, respectively.

## 4. Discussion

Kiwifruit is rich in nutrients, making it susceptible to infection by various pathogens during growth, storage, and transportation. As a result, the appearance and taste of the fruits are seriously affected, and the reduction in fruit quality can lead to huge economic losses [42]. With improved quality of life, consumers are paying more attention to food safety. Guizhou is one of the main production areas of kiwifruit in China, and it is highly important to develop friendly and safe kiwifruit disease control methods to meet the increasing demand for high-quality pollution-free kiwifruit products. *Bacillus* spp. have been widely recognized for their excellent performance in biological control of various plant diseases [43,44].

Previous studies have shown that *B. mojavensis* had inhibitory effects on a variety of pathogens. Galitskaya et al. [45] reported that *B. mojavensis* P1709 could protect postharvest cherry tomatoes from fungal pathogens, and Bacon et al. [46] reported that *Bacillus mojavensis* could reduce stalk lesions in maize seedlings. In this study, *B. mojavensis* BQ-33 had an inhibitory activity on *D. glomerata*, *Botryosphaeria dothidea*, *Alternaria alternata*, *Fusarium oxysporum* and *Phomopsis cauloides*. The in vivo experiment showed that SS with 400 mL L^−1^ had the best control effect on kiwifruit black spot disease, with a control efficacy of 100%. These results indicated that *B. mojavensis* is a promising biocontrol agent in agriculture.

It has been shown that chitinase and β-1,3-glucanase secreted by microorganisms may be key players in inhibiting the growth of fungal pathogens [47]. In this study, the high activities of chitinase and β-1,3-glucanases, which are the key hydrolase enzymes for lysing the cell wall of fungal pathogens, were detected in SS of BQ-33 [48,49]. Chitinase can hydrolyze β-1,4-glycosidic bond in chitin, causing the degradation of the fungal cell wall, while β-1,3-glucanases can catalyze the cleavage of β-1,3-linkages of β-1,3-glucan and show antifungal activity [50,51]. Therefore, BQ-33 may destroy the cell wall of fungal pathogens by producing these two enzymes, thus, inhibiting the growth of fungal pathogens. In the study of Bacon et al. [52], a surfactin was found in the metabolites of *B. mojavensis*, which also had a good antifungal activity. It is plausible that surfactin also has a role in inhibiting pathogens in SS of BQ-33, but this should be further verified.

The cell membrane is a double-layer structure of phospholipids surrounding the cytoplasm. Its main components are lipids and protein molecules, which can provide a protective barrier for small ions (Na^+^, K^+^), macromolecular substances and other cell contents [53,54]. The mycelia of *D. glomerata* treated with SS produced red fluorescence by PI staining, which indicated that SS may damage the integrity of the mycelial cell membrane. Previous studies have shown that when the cell membrane is damaged, it leads to the increase of relative conductivity and the leakage of a large number of intracellular substances [55,56]. In this experiment, the relative conductivity of mycelium medium treated with SS increased significantly, and the leakage of soluble protein and nucleic acid also showed an abnormal upward trend, which was consistent with the above conclusions. These results indicated that SS produced by BQ-33 may also destroy the cell membrane of *D. glomerata*.

The SS produced by BQ-33 displayed greater antagonistic activity against *D. glomerata*, both in vivo and in vitro, which was manifested by the inhibition of colony growth, destruction of cell integrity, and the reduction of disease morbidity of the fruit. Overall, this study shows that the *B. mojavensis* strain, BQ-33, isolated from the rhizosphere soil of kiwifruit plant, displays excellent antifungal activity against fungal pathogens of kiwifruit and has a high potential for use as a biocontrol agent.

## Figures and Tables

**Figure 1 microorganisms-10-02085-f001:**
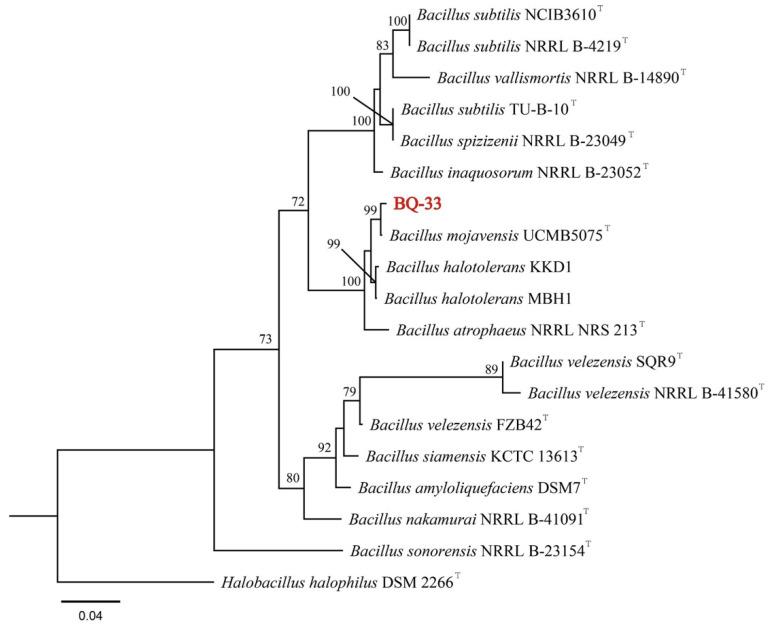
Phylogenetic tree of *Bacillus mojavensis* BQ-33 based on 16S rDNA and *gyrA* tandem gene sequences, constructed using the maximum likelihood (ML) method with MEGA 7.0 software. Bootstrap values are based on 1000 repeats. T = type strain.

**Figure 2 microorganisms-10-02085-f002:**
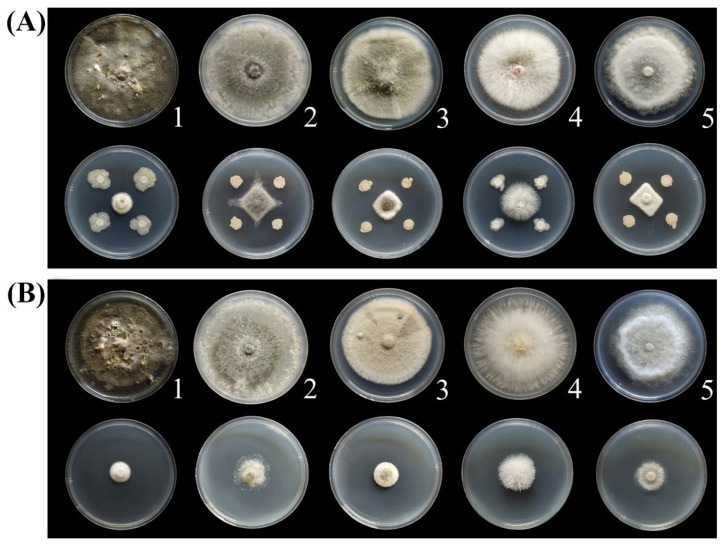
Inhibitory effect of BQ-33 suspension (**A**) and 200 mL L^−1^ sterile supernatant (SS) (**B**) on fungal pathogens. (1) *Didymella glomerata*, (2) *Botryosphaeria dothidea*, (3) *Alternaria alternata*, (4) *Fusarium oxysporum* and (5) *Phomopsis cauloides*.

**Figure 3 microorganisms-10-02085-f003:**
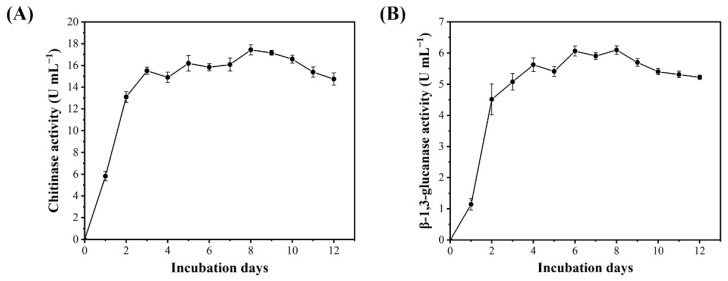
Dynamics in the activities of chitinase (**A**) and β-1,3-glucanase (**B**) in SS of BQ-33 within 12 days.

**Figure 4 microorganisms-10-02085-f004:**
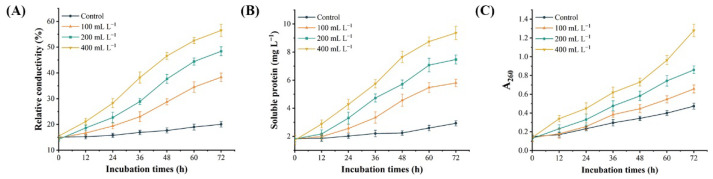
Dynamic changes of relative conductivity (**A**), soluble protein (**B**), and nucleic acid (**C**) in *Didymella glomerata* culture medium treated with SS for 72 h.

**Figure 5 microorganisms-10-02085-f005:**
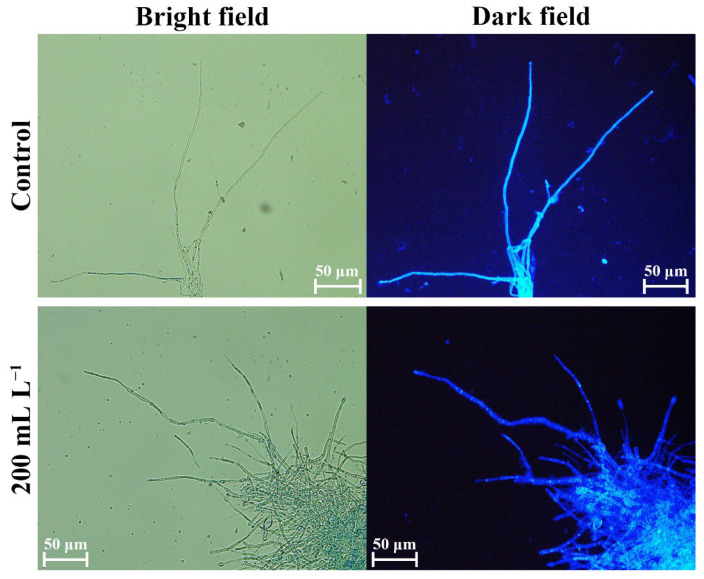
The effect of SS on *D. glomerata* cell wall observed with calcofluor white staining.

**Figure 6 microorganisms-10-02085-f006:**
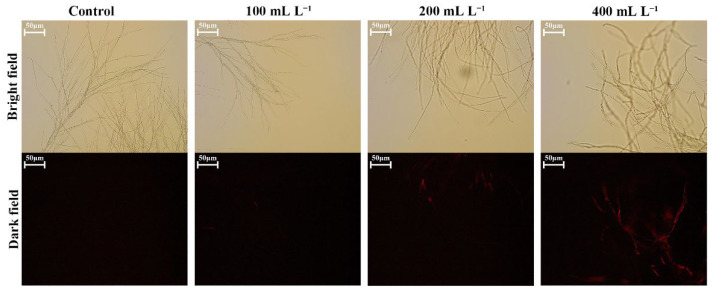
The Effect of SS with different concentrations on the cell membrane permeability of mycelia of *D. glomerata*, as demonstrated by propidium iodide staining.

**Figure 7 microorganisms-10-02085-f007:**
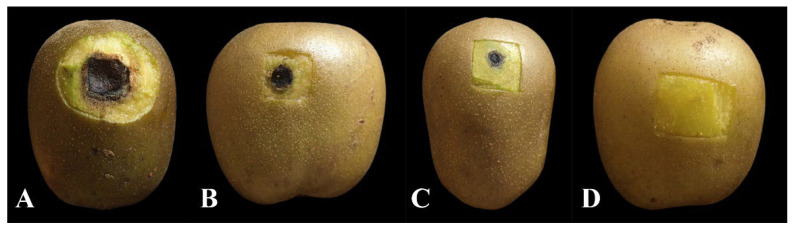
In vivo effects of SFB on kiwifruit disease caused by *D. glomerata*. Infected kiwifruit treated with sterile water (**A**). 100 mL L^−^^1^ SS (**B**). 200 mL L^−^^1^ SS (**C**). 400 mL L^−^^1^ SS (**D**).

**Table 1 microorganisms-10-02085-t001:** The pathogens used in antifungal activity test.

No.	Pathogen	Host Plant	Disease Type
1	*Didymella glomerata*	Kiwifriut	Black spot [6]
2	*Botryosphaeria dothidea*	Apple	Ring rot [31]
3	*Alternaria alternata*	*Prunus salicina*	Leaf spot [32]
4	*Fusarium oxysporum*	*Gastrodia elata*	Tuber rot [33]
5	*Phomopsis cauloides*	Kiwifriut	Soft rot [34]

**Table 2 microorganisms-10-02085-t002:** Inhibitory ratio of QB-33 and its SS against fungal pathogens.

Pathogen	Inhibition Ratio by BQ-33 Suspension (%)	Inhibition Ratio by SS (%)
*Didymella glomerata*	81.26 ± 0.47 a	83.43 ± 0.35 a
*Botryosphaeria dothidea*	73.62 ± 1.43 c	74.19 ± 0.59 c
*Alternaria alternata*	77.45 ± 1.18 b	78.86 ± 0.55 b
*Fusarium oxysporum*	70.11 ± 0.92 d	72.63 ± 0.56 c
*Phomopsis cauloides*	75.73 ± 0.78 bc	77.13 ± 0.67 b

Numerical values were expressed as mean ± standard error (SE) of triplicates. Different lowercase letters represent a significant difference of the same columns (*p <* 0.05, n = 3).

**Table 3 microorganisms-10-02085-t003:** Control efficacy of different concentrations of SS on kiwifruit black spot.

Treatment	Lesion Diameter (mm)	Control Efficacy (%)
Control	20.62 ± 0.66 a	-
100 mL L^−1^	10.62 ± 0.29 b	47.52 ± 0.02 c
200 mL L^−1^	5.12 ± 0.13 c	74.75 ± 0.01 b
400 mL L^−1^	0 ± 0 d	100 ± 0 a

Numerical values were expressed as mean ± SE of triplicates. Different lowercase letters represent a significant difference of the same columns (*p* < 0.05, n = 3).

## Data Availability

The data analyzed in this study are included within the paper.

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
