# Peer review of "Antifungal Activities of Bacillus mojavensis BQ-33 towards the Kiwifruit Black Spot Disease Caused by the Fungal Pathogen Didymella glomerata"

_microorganisms, 2022, doi:10.3390/microorganisms10102085_

Round 1
Reviewer 1 Report
This is a well-written manuscript and provides important information on biological control of D. glomerata associated with kiwifruit black spot disease.
I have no major issues with the submitted manuscript and recommend it accepted for publication. However, I suggest the authors consider the comments listed below prior to final acceptance.
Line 23: Italicize ''in vivo''
Line 23: L-1 instead of L-1
Line 94: Rephrase as ''Petri dishes inoculated only with mycelial plugs were used as controls''
Line 116: OD600 instead of OD600
Line 123: add a comma after ''was tested''
Line 296: Italicize ''Bacillus''
Discussion: My only issue is the Discussion Section of this manuscript and the authors have to elaborate on this part of the manuscript. The length of Discussion is rather short, especially compared with the other sections of the manuscript which are more extensive.

Author Response
List of Responses
Dear Mr. Randal Zhang, Reviewers and Managing Editor
Thank you for your letter and for the reviewers’ comments concerning our manuscript entitled “Antifungal activities of Bacillus mojavensis BQ-33 towards the kiwifruit black spot disease caused by the fungal pathogen Didymella glomerata” (ID: microorganisms-1974276). The comments are all valuable for increasing the significance of our research and very helpful for improving our manuscript. We have made modifications thoroughly according to the comments and we hope that the revisions meet with approval. Revised contents are marked in red in the manuscript. Our responds to the reviewer’s comments are as following:
Responds to the reviewer’s comments:
- Question: Line 23: Italicize “in vivo”
Response: We regret this mistake. In the modified manuscript, the “in vivo” has been marked in italics.
- Question: Line 23: L-1 instead of L-1
Response: We regret this mistake. In the modified manuscript, the “-1” has been superscripted.
- Question: Line 94: Rephrase as “Petri dishes inoculated only with mycelial plugs were used as controls”
Response: We are grateful for the suggestion. In the modified manuscript, we changed “Only inoculated with fungal pathogen disks were used as control.” to “Petri dishes inoculated only with mycelial plugs were used as controls.”.
- Question: Line 116: OD600 instead of OD600
Response: We regret this mistake. In the modified manuscript, the “600” has been subscripted.
- Question: Line 123: add a comma after “was tested”
Response: We are grateful for the suggestion. In the modified manuscript, we have rewritten this passage.
- Question: Line 296: Italicize “Bacillus”
Response: We regret this mistake. In the modified manuscript, the “Bacillus” has been marked in italics.
- Question: Discussion: My only issue is the Discussion Section of this manuscript and the authors have to elaborate on this part of the manuscript. The length of Discussion is rather short, especially compared with the other sections of the manuscript which are more extensive.
Response: We are grateful for the suggestion. In the modified manuscript, we have rewritten the “Discussion” section and added references to fully discuss the results in our study (L342-373).
As mentioned above, we tried our best to improve the manuscript and made some changes in the revised manuscript. These changes will not influence the content and framework of the manuscript.
We appreciate for Mr. Randal Zhang, reviewers and editors' warm work earnestly, and hope that the correction will meet with approval.
Once again, thank you very much for your comments and suggestions.

Reviewer 2 Report
In this study, author isolated a bacterial strain BQ-33 from the rhizosphere soil of kiwifruit. Based on phylogenetic tree constructed using 16S rDNA and the gyrA primers sequences, BQ-33 was identified as Bacillus mojavensis. Confrontation experiments showed that both BQ-33 and the sterile fermentation broth produced by BQ-33 possessed excellent broad-spectrum antifungal activity. Furthermore, the sterile fermentation broth (SFB) damaged the cell membrane and cell wall of the mycelia, resulting in the leakage of a large amount of intracellular material. Chitinase and β-1,3-glucanase activities in SFB increased in correlation with fermentation time and remained at a high level of activity. The article is not well organized. experimental methods lack references. Finally, there are some essential problems should be addressed by authors, which are listed below.
1. Some sentences in the text are not fluent in logic and the wording is not accurate. Suggest reorganizing the language.
2. L19-20 Abbreviations “SFB” may be marked where it first appeared.
3. L23 Does “-1” need to be superscripted?
4. L23 Delete “activity”.
5. L13 “during fruit growth, storage, and transportation” and L36 “during growth and storage”
6. L45-49 Add reference.
7. L50-51 “To reduce the occurrence of kiwifruit black spot disease and improve fruit yield, farmers rely on synthetic fungicides.” Please rewriter this sentence.
8. L71 Abbreviations “SFB” can be used directly.
9. L79-81 “The soil adhered to the roots was collected, dried, and ground and 20 g of each pre-treated sample was mixed with 200 mL of sterile water and incubated in a shaker at 37°C and 150 rpm for 2 h.” Rearrange this sentence.
10. L103-120 Add reference.
11. L116 How long did the OD600 of BQ-33 grow to 0.8. Does 600 need to be subscripted?
12. L123-124 Please explain why were these five fungal species chosen?
13. L155 Please explain what method was used to collect mycelia.
14. L157 What does “AF” stand for?
15. L172 Add the model and company of the CLSM.
16. “2. Materials and methods” part need to add reference.
17. L191 “Each treatment was repeated three times” instead of “using three replicates”.
18. L202 Where were the soil samples collected? 78 strains of bacterial were isolated and whether all of them were sequenced? BQ33 was identified as Bacillus mojavensis and are there any other strains identified as this species?
19. L213 The sequence length of ON231801 does not match the one in GenBank. The accession number of ON245037 is not available in GenBank.
20. L219 I suggest you change it to “Phylogenetic analysis further confirmed that BQ-33 clustered monophyletically with strains of B. mojavensis UCMB5075.”
21. L220 I suggest you change it to “These results indicated that the isolated strain BQ-33 was identified as B. mojavensis.”
22. L267 Why does Figure 6 have four different treatments and Figure 5 only two?
23. L279 It is suggested to add the pictures of kiwifruit pathogenicity to enhance the persuasion of this experiment.
24. L298 “Bacillus” need to be in italics.
25. L291 The discussion part narrates the content of the result part again, but does not discuss the result.
26. The abstract section is about “'Hongyang' kiwifruit (Actinidia chinensis Planch 'Hongyang') black spot caused by the fungal pathogen Didymella glomerata” (L12) and “BQ-33 displayed strong inhibitory effects against fungal pathogens” (L14), the introduction section is about “Kiwifruit black spot can be caused by different fungal pathogens, including Cladosporium cladosporioides, Diaporthe phaseolorum, Alternaria alternata, and Trichothecium roseum” (L40) and “the biocontrol bacterium Bacillus mojavensis strain BQ-33 showed good inhibitory activity against five main fungal pathogens on kiwifruit, especially D. glomerata” (L70), but in this experiment, Didymella glomerata, Botryosphaeria dothidea, Alternaria alternata, Fusarium oxysporum and Phomopsis cauloides were texted, whether all the five fungal pathogens caused kiwifruit black spot disease. There is no inconsistency in this manuscript, please revise the full text. And why do you choose these five bacteria? Please add.
Author Response
List of Responses
Dear Mr. Randal Zhang, Reviewers and Managing Editor
Thank you for your letter and for the reviewers’ comments concerning our manuscript entitled “Antifungal activities of Bacillus mojavensis BQ-33 towards the kiwifruit black spot disease caused by the fungal pathogen Didymella glomerata” (ID: microorganisms-1974276). The comments are all valuable for increasing the significance of our research and very helpful for improving our manuscript. We have made modifications thoroughly according to the comments and we hope that the revisions meet with approval. Revised contents are marked in red in the manuscript. Our responds to the reviewer’s comments are as following:
Responds to the reviewer’s comments:
- Question: Some sentences in the text are not fluent in logic and the wording is not accurate. Suggest reorganizing the language.
Response: In the modified manuscript, we adjusted some inappropriate statements.
- Question: L19-20 Abbreviations “SFB” may be marked where it first appeared.
Response: Thanks for pointing this out. We have marked “SS” where it first appeared in the abstract.
- Question: L23 Does “-1” need to be superscripted?
Response: We regret this mistake. In the modified manuscript, the “-1” has been superscripted.
- Question: L23 Delete “activity”.
Response: We regret this mistake. In the modified manuscript, the excess “activity” has been removed.
- Question: L13 “during fruit growth, storage, and transportation” and L36 “during growth and storage” have different meanings.
Response: We regret this mistake. In the modified manuscript, we modified these descriptions as “during fruit growth”.
- Question: L45-49 Add reference.
Response: We are grateful for the suggestion. In fact, this description is based on our investigations at some locations in Liupanshui city of Guizhou during past years, but there is not publication as yet. Therefore, we delete this sentence from our manuscript to avoid misunderstandings.
- Question: L50-51 “To reduce the occurrence of kiwifruit black spot disease and improve fruit yield, farmers rely on synthetic fungicides.” Please rewriter this sentence.
Response: We are grateful for the suggestion. In the modified manuscript, we have changed this sentence to “In production, farmers mainly rely on synthetic fungicides to reduce the occurrence of kiwifruit black spots and improve fruit quality”.
- Question: L71 Abbreviations “SFB” can be used directly.
Response: Thanks for pointing this out. In the modified manuscript, we have made the modification throughout the manuscript.
- Question: L79-81 “The soil adhered to the roots was collected, dried, and ground and 20 g of each pre-treated sample was mixed with 200 mL of sterile water and incubated in a shaker at 37°C and 150 rpm for 2 h.” Rearrange this sentence.
Response: We regret this mistake. In the modified manuscript, we have changed this sentence to “From each sample, 20 g of dried and ground soil was mixed with 200 mL of sterile water and incubated in a shaker at 37°C (150 rpm) for 2 h”.
- Question: L103-120 Add reference.
Response: We are grateful for the suggestion. In the modified manuscript, we have added appropriate references to verify the feasibility of our experimental methods.
- Question: L116 How long did the OD600 of BQ-33 grow to 0.8. Does 600 need to be subscripted?
Response: We are grateful for the suggestion. In our experiment, the OD600 of BQ-33 medium usually reached 0.8 after 48h of culture. In the modified manuscript, we have changed the passage to “BQ-33 was inoculated into NB and incubated in a shaking (34°C,180rpm) for 48 h, and the OD600 reached 0.8”, and the "600" has been subscripted.
- Question: L123-124 Please explain why were these five fungal species chosen?
Response: Some pathogens on kiwifruit can also infect other crops and cause diseases. Our original intention is to explore whether Bacillus mojavensis BQ-33 has good inhibitory activity against these pathogens, these results can reflect the broad-spectrum of BQ-33 in agricultural production. We may not have expressed this clearly in the previous manuscript, but we have made a revision in the modified manuscript.
- Question: L155 Please explain what method was used to collect mycelia.
Response: We are grateful for the suggestion. In the modified manuscript, we have added the following sentence to make the description more complete. “three layers of sterile gauze were used to filter and collect the mycelia, and the collected mycelia were washed three times with sterile water”.
- Question: L157 What does “AF” stand for?
Response: We regret this mistake. In the modified manuscript, the “AF” was corrected as the “SS” (sterile supernatant).
- Question: L172 Add the model and company of the confocal laser scanning microscope (CLSM).
Response: We are grateful for the suggestion. In the modified manuscript, we have added the company and model of CLSM.
- Question: “2. Materials and methods” part need to add reference.
Response: We are grateful for the suggestion. In the modified manuscript, we have added appropriate references in the “Materials and Methods”.
- Question: L191 “Each treatment was repeated three times” instead of “using three replicates”.
Response: We are grateful for the suggestion. In the modified manuscript, we changed “using three replicates” to “Each treatment was repeated three times”.
- Question: L202 Where were the soil samples collected? 78 strains of bacterial were isolated and whether all of them were sequenced? BQ33 was identified as Bacillus mojavensis and are there any other strains identified as this species?
Response: We are grateful for the suggestion. In our experiment, we first tested the antifungal effect of bacterial strains isolated from soil. The biocontrol bacterium with the best inhibitory activity against Didymella glomerata was named BQ-33. Then we only sequenced BQ-33 and identified it as Bacillus mojavensis. In addition, we added the source of soil samples to Section 2.2 (L 83).
- Question: L213 The sequence length of ON231801 does not match the one in GenBank. The accession number of ON245037 is not available in GenBank.
Response: We regret this mistake. In the modified manuscript, we corrected the length of the two gene sequences and temporarily removed their accession numbers. In fact, we have received an email from NCBI about the GeneBank accepting BQ-33 sequencing results for gyrA gene, but the accession number of ON245037 can't be found in the GeneBank, which may be because NCBI has not disclosed the sequence. We have sent an email to NCBI, hoping to release the sequence information in advance, and we got the reply that the gene sequence information with the accession number of ON245037 will be released on October 17 th.
- Question: L219 I suggest you change it to “Phylogenetic analysis further confirmed that BQ-33 clustered monophyletically with strains of B. mojavensis UCMB5075.”
Response: We are grateful for the suggestion. In the modified manuscript, we changed “Phylogenetic analysis further confirmed that BQ-33 and Bacillus mojavensis UCMB5075 clustered together” to “Phylogenetic analysis further confirmed that BQ-33 clustered monophyletically with strain of B. mojavensis UCMB5075”.
- Question: L220 I suggest you change it to “These results indicated that the isolated strain BQ-33 was identified as B. mojavensis.”
Response: We are grateful for the suggestion. In the modified manuscript, we changed “These results indicate that the isolated strain BQ-33 is B. mojavensis.” to “Based on these results, the isolated strain BQ-33 was identified as B. mojavensis.”
- Question: L267 Why does Figure 6 have four different treatments and Figure 5 only two?
Response: We are grateful for the suggestion. In the previous experiment (Figure 5), we detected the activities of chitinase and β-1,3-glucanase in SS, and our original intention in CFW staining experiment was just to verify the interruption effect of the middle concentration of these three concentrations on cell wall. The results showed that the interruption effect of SS on cell wall was very obvious, and therefore we didn't test other concentrations.
- Question: L279 It is suggested to add the pictures of kiwifruit pathogenicity to enhance the persuasion of this experiment.
Response: We are grateful for the suggestion. We have added relevant images to the manuscript in the appropriate place.
- Question: L298 “Bacillus” need to be in italics.
Response: We regret this mistake. In the modified manuscript, we have corrected this mistake.
- Question: L291 The discussion part narrates the content of the result part again, but does not discuss the result.
Response: We are grateful for the suggestion. In the modified manuscript, we have rewritten the “Discussion” section and added references to fully discuss the results in our study (L342-373).
- The abstract section is about “'Hongyang' kiwifruit (Actinidia chinensis Planch 'Hongyang') black spot caused by the fungal pathogen Didymella glomerata” (L12) and “BQ-33 displayed strong inhibitory effects against fungal pathogens” (L14), the introduction section is about “Kiwifruit black spot can be caused by different fungal pathogens, including Cladosporium cladosporioides, Diaporthe phaseolorum, Alternaria alternata, and Trichothecium roseum” (L40) and “the biocontrol bacterium Bacillus mojavensis strain BQ-33 showed good inhibitory activity against five main fungal pathogens on kiwifruit, especially D. glomerata” (L70), but in this experiment, Didymella glomerata, Botryosphaeria dothidea, Alternaria alternata, Fusarium oxysporum and Phomopsis cauloides were texted, whether all the five fungal pathogens caused kiwifruit black spot disease. There is no inconsistency in this manuscript, please revise the full text. And why do you choose these five bacteria? Please add.
Response: We are grateful for the suggestion. In the modified manuscript, we described in detail why we chose these five fungal pathogens.
As mentioned above, we tried our best to improve the manuscript and made some changes in the revised manuscript. These changes will not influence the content and framework of the manuscript.
We appreciate for Mr. Randal Zhang, reviewers and editors' warm work earnestly, and hope that the correction will meet with approval.
Once again, thank you very much for your comments and suggestions.

Reviewer 3 Report
The study by Wang et al. for first time evaluated the biological control of kiwifruit black spots caused by Didymella glomerata. Exploring alternative measures to control different plant pathogens is of immense importance in light of the finding substitutes for harmful chemical pesticides. The topic of the study is significant, and various methods were used to characterize the biocontrol potential of Bacillus mojavensis strain isolated from the kiwifruit rhizosphere. The Methods and Results sections need to be improved according to the comments bellow. Also, the discussion is very short and needs to be broadened.
Some comments that also should be considered:
Line 11: add cultivar or variety before 'Hongyang'
Line 13: Change “discover” to “identify”
Line 14: change sentence “In this study, bacterial strain BQ-33, which displayed strong inhibitory effects against fungal pathogens, was isolated from the rhizosphere soil of kiwifruit” to “In this study, bacterial isolates from the rhizosphere soil of kiwifruit were tested for their potential antifungal activity against selected fungal pathogens.”
Line 16: Please change the sentence to "BQ-33 isolate displayed strongest inhibitory effect and according to 16S rDNA and the gyrA gene sequences, was identified as Bacillus mojavensis."
Line 21: which intracellular material, specify
Line 22: change “fermentation” to “incubation”
Line 80: what is pre-treated sample?
Line 83: delete “to a series of concentrations” and merge the sentence with the next one
Line 91: has it been tested overnight culture of BQ-33, or diluted suspension?
Line 94: add “plates” before “inoculated”
Line 99: this is a result, you can say that the best candidate was further identified according to 16S rDNA and gyrB sequence
Line 100-102: remove the sentence, it is redundant
Line 102: begin a sentence with “For molecular identification”
Line 109: remove “those in”
Line 111: use “reference strain sequences” instead of “associate sequences”
Line 115: change the title according to the comment on line 118
Line 118: it should be avoided the use of the term “fermentation broth” considering that a bioreactor was not used for the preparation of culture, but instead was routinely prepared in a shaker incubator. So, it can be used term “supernatant”. Change it throughout the manuscript.
Line 121: change the title to “Test of antifungal activity of BQ-33 supernatant”
Line 123: were these pathogens previously identified?
Line 129: How percent of mycelial growth inhibition was calculated?
Line 138: remove “was added”
Line 153: reference for method?
Line 157: to compare changes in conductivity, proteins and nucleic acids in treatment there is a need for control only with D. glomerata mycelia – or it was only PDB medium used as control?
Line 157: what is AF?
Line 159: How the relative conductivity was calculated?
Line 167: how the concentration of 200 mL L-1 was selected for further tests
Line 160: change “protein was used” to “protein was detected”
Line 164: reference for method? Also, rephrase the sentence “The cell walls of D. glomerata were stained with calcofluor white (CFW)” to “Calcofluor white (CFW) was used to detect changes in the integrity of cell walls of D. glomerata”
Line 174: reference for method?
Line 203: where is Table S1? No supplementary data is available for review
Line 227: from this subsection please refer to the strain BQ-33 as Bacillus mojavensis BQ-33
Line 227: emphasize that tested overnight culture (or suspension, depending on answer to line 91) of B. mojavensis BQ-33 and its SBF, and further where it appears in the manuscript
Line 228: change “on the fruits, leaves, and branches of kiwifruit plants” to “of the fruits, leaves, and branches of kiwifruit plants”
Line 230: change “morphologically abnormal colonies” to significantly reduced mycelial growth
Line 256: change “improved” to “affected”
Line 259: how it was concluded that D. glomerata was unable to grow normally. Do the appropriate aliquots during the experiment were plated to monitor fungal growth? Or optical density of fungal growth was measured?
Line 281: in the method section was explained that kiwifruit was sprayed with SBF, and fungal inocula were somewhere in the box. It implies that kiwifruit was not infected at the begining of the experiment. Infection occurred during incubation. So, change “different concentrations of SFB were used to treat kiwifruit fruits infected with D. glomerata” to “different concentrations of supernatant were tested on kiwifruit.”
Line 287: please add a picture of the biocontrol results
Line 291: The discussion is very short, and needs to be broadened with more literature data.
Line 300: change “on” to “against”
Line 301-302: if the BQ-33 was firstly tested in this study then you cannot use this statement. Instead, discuss antifungal Bacillus mojavensis from previous studies and compare your results
Author Response
List of Responses
Dear Mr. Randal Zhang, Reviewers and Managing Editor
Thank you for your letter and for the reviewers’ comments concerning our manuscript entitled “Antifungal activities of Bacillus mojavensis BQ-33 towards the kiwifruit black spot disease caused by the fungal pathogen Didymella glomerata” (ID: microorganisms-1974276). The comments are all valuable for increasing the significance of our research and very helpful for improving our manuscript. We have made modifications thoroughly according to the comments and we hope that the revisions meet with approval. Revised contents are marked in red in the manuscript. Our responds to the reviewer’s comments are as following:
Responds to the reviewer’s comments:
- Question: Line 11: add cultivar or variety before 'Hongyang'
Response: We are grateful for the suggestion. In the modified manuscript, we changed “Actinidia chinensis Planch 'Hongyang'” to “Actinidia chinensi, cultivar 'Hongyang'”.
- Question: Line 13: Change “discover” to “identify”
Response: We are grateful for the suggestion. In the modified manuscript, we changed “discover” to “identify”.
- Question: Line 14: change sentence “In this study, bacterial strain BQ-33, which displayed strong inhibitory effects against fungal pathogens, was isolated from the rhizosphere soil of kiwifruit” to “In this study, bacterial isolates from the rhizosphere soil of kiwifruit were tested for their potential antifungal activity against selected fungal pathogens.”
Response: We are grateful for the suggestion. In the modified manuscript, we changed “In this study, bacterial strain BQ-33, which displayed strong inhibitory effects against fungal pathogens, was isolated from the rhizosphere soil of kiwifruit” to “In this study, bacterial isolates from the rhizosphere soil of kiwifruit were tested for their potential antifungal activity against selected fungal pathogens.”.
- Question: Line 16: Please change the sentence to “BQ-33 isolate displayed strongest inhibitory effect and according to 16S rDNA and the gyrA gene sequences, was identified as Bacillus mojavensis.”
Response: We are grateful for the suggestion. In the modified manuscript, we changed “Based on a phylogenetic tree constructed using sequences of 16S rDNA and the gyrA gene, BQ-33 was identified as Bacillus mojavensis.” to “BQ-33 isolate displayed strongest inhibitory effect and according to 16S rDNA and the gyrA gene sequences, was identified as Bacillus mojavensis.”
- Question: Line 21: which intracellular material, specify
Response: We are grateful for the suggestion. In the modified manuscript, we list leaked intracellular materials such as small ions (Na+, K+), soluble proteins and nucleic acids.
- Question: Line 22: change “fermentation” to “incubation”
Response: We are grateful for the suggestion. In the modified manuscript, we changed “fermentation” to “incubation”.
- Question: Line 80: what is pre-treated sample?
Response: We regret this mistake. In the modified manuscript, we redescribe this part of the content. We have changed this sentence to “From each sample, 20 g of dried and ground soil was mixed with 200 mL of sterile water and incubated in a shaker at 37°C (150 rpm) for 2 h”.
- Question: Line 83: delete “to a series of concentrations” and merge the sentence with the next one
Response: We are grateful for the suggestion. In the modified manuscript, we changed “the supernatant was diluted to a series of concentrations. Next, 100 μL of the 10-3, 10-4, and 10-5 dilutions was spread on nutrient agar (NA) medium” to “100 μL of the supernatant diluted to 10-3, 10-4, and 10-5 was spread on nutrient agar (NA) medium”.
- Question: Line 91: has it been tested overnight culture of BQ-33, or diluted suspension?
Response: We are grateful for the suggestion. In the modified manuscript, we changed “Suspensions of antagonistic bacteria were applied to sterilised filter paper disks” to “Antagonistic bacteria suspension incubated overnight were applied to sterilised filter paper disks”
- Question: Line 94: add “plates” before “inoculated”
Response: We are grateful for the suggestion. In the modified manuscript, we changed “Only inoculated with fungal pathogen disks were used as control.” to “Only plates inoculated with fungal pathogen disks were used as control.”
- Question: Line 99: this is a result, you can say that the best candidate was further identified according to 16S rDNA and gyrB sequence
Response: We are grateful for the suggestion. In the modified manuscript, we changed “Evaluation of antifungal activity showed that the bacterial strain BQ-33 displayed the strongest inhibitory effect on D. glomerata.” to “The best candidate was further identified according to 16S rDNA and gyrA sequence.”.
- Question: Line 100-102: remove the sentence, it is redundant
Response: We are grateful for the suggestion. In the modified manuscript, we have removed this sentence.
- Question: Line 102: begin a sentence with “For molecular identification”
Response: We are grateful for the suggestion. In the modified manuscript, we changed “For this” to “For molecular identification”.
- Question: Line 109: remove “those in”
Response: We are grateful for the suggestion. In the modified manuscript, we have removed “those in”.
- Question: Line 111: use “reference strain sequences” instead of “associate sequences”
Response: We are grateful for the suggestion. In the modified manuscript, we changed “associate sequences” to “reference strain sequences”.
- Question: Line 115: change the title according to the comment on line 118
Response: We are grateful for the suggestion. In the modified manuscript, we changed “Preparation of QB-33 The sterile fermentation broth (SFB)” to “Preparation of QB-33 the sterile supernatant (SS)”
- Question: Line 118: it should be avoided the use of the term “fermentation broth” considering that a bioreactor was not used for the preparation of culture, but instead was routinely prepared in a shaker incubator. So, it can be used term “supernatant”. Change it throughout the manuscript.
Response: We are grateful for the suggestion. We changed all the “fermentation broth” in the manuscript to “supernatant”.
- Question: Line 121: change the title to “Test of antifungal activity of BQ-33 supernatant”
Response: We are grateful for the suggestion. Because this part of the experiment includes the antifungal activity of BQ-33 and its sterile supernatant, we changed the title to “Test of antifungal activity of BQ-33 and its SS.”
- Question: Line 123: were these pathogens previously identified?
Response: We are grateful for the suggestion. In the modified manuscript, the selected pathogenic fungi are listed in Table 1, with references.
- Question: Line 129: How percent of mycelial growth inhibition was calculated?
Response: We are grateful for the suggestion. In the modified manuscript, we add the calculation of the inhibition ratio in Section 2.5
- Question: Line 138: remove “was added”
Response: We are grateful for the suggestion. In the modified manuscript, we have removed “was added”.
- Question: Line 153: reference for method?
Response: We regret this mistake. In the modified manuscript, we have added appropriate references to verify the feasibility of our experimental methods
- Question: Line 157: to compare changes in conductivity, proteins and nucleic acids in treatment there is a need for control only with D. glomerata mycelia – or it was only PDB medium used as control?
Response: We are grateful for the suggestion. In the modified manuscript, we changed “PDB without AF was used as a control.” to “In the control, mycelia were incubated in PDB without SS”
- Question: Line 157: what is AF?
Response: We are grateful for the suggestion. In the modified manuscript, the “AF” was corrected as the “SS”.
- Question: Line 159: How the relative conductivity was calculated?
Response: We regret this mistake. We calculated the relative conductivity according to the method of Mo et al. In the modified manuscript, we redescribed this sentence and added references.
- Question: Line 167: how the concentration of 200 mL L-1 was selected for further tests
Response: In the previous experiment (Figure 5), we detected the activities of chitinase and β-1,3-glucanase in SFB, and our original intention in CFW staining experiment was just to verify the interruption effect of the middle concentration of these three concentrations on cell wall. The results showed that the interruption effect of SFB on cell wall was very obvious, and therefore we didn't test other concentrations.
- Question: Line 160: change “protein was used” to “protein was detected”
Response: We are grateful for the suggestion. In the modified manuscript, we changed “For this” to “For molecular identification”.
- Question: Line 164: reference for method? Also, rephrase the sentence “The cell walls of D. glomerata were stained with calcofluor white (CFW)” to “Calcofluor white (CFW) was used to detect changes in the integrity of cell walls of D. glomerata”
Response: We are grateful for the suggestion. In the modified manuscript, we changed “The cell walls of D. glomerata were stained with calcofluor white (CFW)” to “Calcofluor white (CFW) was used to detect changes in the integrity of cell walls of D. glomerata”, and added references to the method.
- Question: Line 174: reference for method?
Response: We are grateful for the suggestion. In the modified manuscript, we have added appropriate references in the “Materials and Methods”.
- Question: Line 203: where is Table S1? No supplementary data is available for review
Response: We regret this negligence. A supplementary file containing Table S1 have been submitted together with the revised manuscript.
- Question: Line 227: from this subsection please refer to the strain BQ-33 as Bacillus mojavensis BQ-33
Response: We are grateful for the suggestion. In Section 3.1, we change “BQ-33” to “B. mojavensis BQ-33”.
- Question: Line 227: emphasize that tested overnight culture (or suspension, depending on answer to line 91) of B. mojavensis BQ-33 and its SBF, and further where it appears in the manuscript
Response: We are grateful for the suggestion. In the modified manuscript, we change this type of error to “BQ-33 suspension”.
- Question: Line 228: change “on the fruits, leaves, and branches of kiwifruit plants” to “of the fruits, leaves, and branches of kiwifruit plants”
Response: We are grateful for the suggestion. In the modified manuscript, we change “on the fruits, leaves, and branches of kiwifruit plants” to “of the fruits, leaves, and branches of kiwifruit plants”.
- Question: Line 230: change “morphologically abnormal colonies” to significantly reduced mycelial growth
Response: We are grateful for the suggestion. In the modified manuscript, we change “morphologically abnormal colonies” to “significantly reduced mycelial growth”.
- Question: Line 256: change “improved” to “affected”
Response: We are grateful for the suggestion. In the modified manuscript, we change “improved” to “affected”.
- Question: Line 259: how it was concluded that D. glomerata was unable to grow normally. Do the appropriate aliquots during the experiment were plated to monitor fungal growth? Or optical density of fungal growth was measured?
Response: In Figure 2B (1), compared with the control group, the growth of D. glomerata in PDA containing SS was inhibited, so it could not grow normally.
- Question: Line 281: in the method section was explained that kiwifruit was sprayed with SBF, and fungal inocula were somewhere in the box. It implies that kiwifruit was not infected at the begining of the experiment. Infection occurred during incubation. So, change “different concentrations of SFB were used to treat kiwifruit fruits infected with D. glomerata” to “different concentrations of supernatant were tested on kiwifruit.”
Response: We are grateful for the suggestion. In the modified manuscript, we change “different concentrations of SFB were used to treat kiwifruit fruits infected with D. glomerata” to “different concentrations of supernatant were tested on kiwifruit.”.
- Question: Line 287: please add a picture of the biocontrol results
Response: We are grateful for the suggestion. We have added relevant images to the manuscript in the appropriate place.
- Question: Line 291: The discussion is very short, and needs to be broadened with more literature data.
Response: We are grateful for the suggestion. In the modified manuscript, we have rewritten the “Discussion” section and added references to fully discuss the results in our study (L342-373).
- Question: Line 300: change “on” to “against”
Response: We are grateful for the suggestion. In the modified manuscript, we have rewritten this passage.
- Question: Line 301-302: if the BQ-33 was firstly tested in this study then you cannot use this statement. Instead, discuss antifungal Bacillus mojavensis from previous studies and compare your results
Response: We are grateful for the suggestion. In the modified manuscript, we have rewritten this passage.
As mentioned above, we tried our best to improve the manuscript and made some changes in the revised manuscript. These changes will not influence the content and framework of the manuscript.
We appreciate for Mr. Randal Zhang, reviewers and editors' warm work earnestly, and hope that the correction will meet with approval.
Once again, thank you very much for your comments and suggestions.

Reviewer 4 Report
Didymella glomerata has been identified as the main pathogen causing black spot on ‘Hongyang’ kiwifruit. In addition, it can infect kiwifruits at different stages of fruit growth and during postharvest storage. Farmers commonly use chemical fungicides because they can quickly and effectively reduce the severity of fungal disease, but intensive use of fungicides can accelerate the development of resistance in fungal pathogens. In addition, fungicides contaminate the environment and food. These are the reasons why the search for safe antifungal agents is so important.
Biocontrol of fungal diseases by certain species of bacteria or symbiotic fungi, is an alternative to synthetic fungicides.
In this study Authors analyze antifungal properties of Bacillus mojavensis BQ-33 and its metabolites against five main fungal pathogens on kiwifruit, especially Didymella glomerata. The strain of Bacillus was isolated from rhizosphere soil from healthy fruit trees in kiwifruit orchards.
The Authors isolated 78 bacterial strains and among them 9 strains with more than 70% inhibition ratio against Didymella. The BQ-33 strain remains the best in the inhibition ration (81.62%). Antifungal properties were also demonstrated for 4 other fungal pathogens and it turned out that the growth of all pathogens was significantly inhibited. I consider this research valuable.
However, I have minor comments on the presented results and on the discussion
Figure 2
control of Bacillus grown on PDA medium should be added
Control growth of pathogenic fungi on PDA is the same for both experiments
The figure should have only three rows
1. Control – pathogenic fungi on PDA medium
2. inhibitory effect of BQ-33 - pathogenic fungi in the center of the plates and 4 disks soaked with BQ-33 located around
3. inhibitory effect of SFB pathogenic fungi grown on the PDA medium supplemented with SFB
the inhibitory effect of Bacillus BQ-33 (panel A) is clear for Botryosphaeria, Altrnaria and Phomopsis but the discs with bacteria look different for Didymella (1) and Fusarium (4). For Didymella something is growing around the four discs (what is that?) and for Fusarium the Fusarium colony looks round and probably should be grown longer to show the inhibitory effect. Fungal colonies do not have distinct zones of growth inhibition in the form of indentation such as in Botryosphaeria, Altrnaria and Phomopsis
SBF was collected after 4 days when activity of hydrolytic enzymes reached maximum. The Authors looking for mechanisms of antifungal activity of BQ-33 examined the hydrolytic consequences of SFB added to the culture medium of Didymella.
It is interesting to use SFB to combat fungal diseases, however, problems may be encountered with such storage of SFB as not to lose its hydrolytic activity. The Authors showed that this activity gradually declines slowly after 8 days. Do the Authors try to store the SFB in some protective conditions?
Discussion should be enriched.
The discussion is essentially a summary of the results.
The Authors demonstrated high activity of hydrolytic enzymes in the culture medium (SFB) and suggested that hydrolases play an important role in the antifungal activity of BQ33. Much information can be found in the literature on various Bacillus strains that promote plant growth and exhibit antifungal properties. Bacillus mojavensis produces sulfactin with antifungal activity (Bacon et al. 2012). Is there anything known about the synergistic effects of sulfactanes and hydrolytic enzymes? The authors use SFB, which may contain both compounds and other unknown metabolites.
However, the in vitro inhibition observed by Bacon et al. (2012) did not necessarily refer to total surfactin concentration, suggesting a complex mechanism of inhibition and / or the presence of other unknown factors. This observation underscores the importance of other factors in the antifungal activity and hydrolytic enzymes may play a role here. The Authors found two publications (34, 35) on the antifungal activity of B. mojavensis and it is an opportunity to discuss with them.
Abstract line 23 remove please one “activity”
Line 203 Table S1 – not available
Discussion line 302 – antibacterial or antifungal ????
Author Response
List of Responses
Dear Mr. Randal Zhang, Reviewers and Managing Editor
Thank you for your letter and for the reviewers’ comments concerning our manuscript entitled “Antifungal activities of Bacillus mojavensis BQ-33 towards the kiwifruit black spot disease caused by the fungal pathogen Didymella glomerata” (ID: microorganisms-1974276). The comments are all valuable for increasing the significance of our research and very helpful for improving our manuscript. We have made modifications thoroughly according to the comments and we hope that the revisions meet with approval. Revised contents are marked in red in the manuscript. Our responds to the reviewer’s comments are as following:
Responds to the reviewer’s comments:
- Question: In Figure 2. The control of Bacillus grown on PDA medium should be added.
Response: We are grateful for the suggestion. In fact, in the measurement of antifungal activity in vitro, there is no strict requirement to show that only biocontrol bacteria grown on PDA. For reference, we listed two articles in the field of biological control published in recent five years.
Reference
- Zhang, F.; Li, X.; Zhu, S.; Ojaghian. M.R.; Zhang, J. Biocontrol potential of Paenibacillus polymyxa against Verticillium dahlia infecting cotton plants. Control 2018, 127, 70-77. https://doi.org/10.1016/j.biocontrol.2018.08.021
- Cui, W.; He, P.; Munir, S.; He, P.; Li, X.; Li, Y.; Wu, J.; Wu, Y.; Yang, L.; He, P.; He, Y. Efficacy of plant growth promoting bacteria Bacillus amyloliquefaciens B9601-Y2 for biocontrol of southern corn leaf blight. Control 2019, 139, 104080. https://doi.org/10.1016/j.biocontrol.2019.104080
- Question: In Figure 2. The figure should have only three rows: (1) Control – pathogenic fungi on PDA medium. (2) inhibitory effect of BQ-33 - pathogenic fungi in the center of the plates and 4 disks soaked with BQ-33 located around. (3) inhibitory effect of SFB pathogenic fungi grown on the PDA medium supplemented with SFB.
Response: We are grateful for the suggestion. Figure 2 A and B are separately conducted experiments. In this evaluation, we first tested the inhibitory activity of BQ-33 against the five pathogenic bacteria. After we found that BQ-33 could effectively inhibit these five pathogenic bacteria, we further chose sterile supernatant (SS) for the experiment. In figure 2B, the control medium has been modified with the same amount of NB medium that are used for fermentation. Therefore, there are pictures of the control groups.
- Question: the inhibitory effect of Bacillus BQ-33 (panel A) is clear for Botryosphaeria, Altrnaria and Phomopsis but the discs with bacteria look different for Didymella (1) and Fusarium (4). For Didymella something is growing around the four discs (what is that?) and for Fusarium the Fusarium colony looks round and probably should be grown longer to show the inhibitory effect. Fungal colonies do not have distinct zones of growth inhibition in the form of indentation such as in Botryosphaeria, Altrnaria and Phomopsis.
Response: We are grateful for the suggestion. In fact, BQ-33 glomerata showed very good inhibition effect on Didymella glomerata. The reason why Didymella glomerata colonies are still round may be that some substance secreted by BQ-33 could inhibit the growth of Didymella glomerata colonies very well from the beginning, resulting in very slow growth of this pathogen. On the other hand, some bioactive substances secreted by BQ-33 against Didymella glomerata may diffuse easily in PDA, so they can uniformly inhibit the growth of Didymella glomerata. The inhibitory effect of BQ-33 on Fusarium oxysporum was lower than that of the other four pathogens, as shown in Table 2. There were obvious traces of inhibition when Fusarium oxysporum was exposed to PDA, however, dense aerial hyphae of Fusarium oxysporum could still grow without contacting PDA, and it appeared that the treatment group of Fusarium oxysporum were different from other pathogens.
Around the four filter papers is the colonies of biocontrol bacterium BQ-33. When the filter paper soaked with BQ-33 bacteria suspension was inoculated onto PDA, the bacteria would slowly spread around after contacting the nutrients. The interaction between biocontrol bacteria and different pathogenic bacteria may be the reason for the difference of biocontrol bacteria colony size.
- Question: It is interesting to use SFB to combat fungal diseases, however, problems may be encountered with such storage of SFB as not to lose its hydrolytic activity. The Authors showed that this activity gradually declines slowly after 8 days. Do the authors try to store the SFB in some protective conditions?
Response: We are grateful for the suggestion. In addition to enzymes, there may be many other antifungal substances in the sterile fermentation broth of biocontrol bacteria. In our study, we only made a preliminary exploration of the hydrolase. In the experiment, we usually stored SS at 4℃ for a short period of time (within one week), which showed no difference with fresh SS in antifungal activity. Therefore, we speculate that a low temperature would be a protective condition for the storage of SS.
- Question: Discussion should be enriched.
Response: We are grateful for the suggestion. In the modified manuscript, we have rewritten the “Discussion” section and added references to fully discuss the results in our study (L342-373).
- Question: Abstract line 23 remove please one “activity”
Response: We regret this mistake. In the modified manuscript, the excess “activity” has been removed.
- Question: Line 203 Table S1 – not available
Response: We regret this negligence. A supplementary file containing Table S1 have been submitted together with the revised manuscript.
- Question: Discussion line 302 – antibacterial or antifungal?
Response: We regret this mistake. In the modified manuscript, we have replaced all the “antibacterial” with “antifungal”.
As mentioned above, we tried our best to improve the manuscript and made some changes in the revised manuscript. These changes will not influence the content and framework of the manuscript.
We appreciate for Mr. Randal Zhang, reviewers and editors' warm work earnestly, and hope that the correction will meet with approval.
Once again, thank you very much for your comments and suggestions.

Reviewer 5 Report
In this study, the antifungal properties of Bacillus mojavensis BQ-33 have been evaluated against kiwifruit pathogen Didymella glomerata. The study contains novelty, the manuscript is well organized and the results are interesting due to the lack of methods to control this pathogen. I have some minor concerns that must be addressed before the manuscript can be considered for publication.
1. The manuscript contains numerous typing and grammar mistakes that must be corrected.
2. There is no information regarding how the pathogen was identified. If the authors reported the isolations and identification of this pathogen previously, the reference must be added.
3. There is lack of information regarding how many repetitions were carried out for each experiment.
4. How the kiwifruit used in the study were manipulated before the antifungal screening? Harvested kiwifruit were used? What was the stage of maturity?
5. Statistical analysis must be added in Figure 4.
6. The in vivo efficacy was screened using only a few conditions. Only curative efficacy was assayed? And preventive efficacy? What was the concentration of SFB (CFU/mL) in the sprayed solution?
7. The discussion must be extended by compering this study with other reported methods.
Author Response
List of Responses
Dear Mr. Randal Zhang, Reviewers and Managing Editor
Thank you for your letter and for the reviewers’ comments concerning our manuscript entitled “Antifungal activities of Bacillus mojavensis BQ-33 towards the kiwifruit black spot disease caused by the fungal pathogen Didymella glomerata” (ID: microorganisms-1974276). The comments are all valuable for increasing the significance of our research and very helpful for improving our manuscript. We have made modifications thoroughly according to the comments and we hope that the revisions meet with approval. Revised contents are marked in red in the manuscript. Our responds to the reviewer’s comments are as following:
Responds to the reviewer’s comments:
- Question: The manuscript contains numerous typing and grammar mistakes that must be corrected.
Response: In the modified manuscript, we adjusted some inappropriate statements.
- Question: There is no information regarding how the pathogen was identified. If the authors reported the isolations and identification of this pathogen previously, the reference must be added.
Response: We are grateful for the suggestion. In the modified manuscript, we have added the source of Didymella glomerata and related references.
- Question: There is lack of information regarding how many repetitions were carried out for each experiment.
Response: We are grateful for the suggestion. In fact, we will conduct pre-experiment before each experiment, then repeat the experiment and record the data. The results of pre-experiment and follow-up experiment are similar.
- Question: How the kiwifruit used in the study were manipulated before the antifungal screening? Harvested kiwifruit were used? What was the stage of maturity?
Response: We are grateful for the suggestion. In the modified manuscript, we added some relevant contents. Healthy kiwifruit (90 days after fruit setting) with consistent maturity were selected in this experiment. Before the in vivo experiment, we preserved kiwifruit at 4℃ for 2 days.
- Question: Statistical analysis must be added in Figure 4.
Response: We are grateful for the suggestion. In most relevant studies, this part of significance analysis does not need to be reflected in the figure. Consider the following three references.
- Wang, Y.; Liu, X.; Chen, T.; Xu, Y.; Tian, S. Antifungal effects of hinokitiol on development of Botrytis cinerea in vitro and in vivo. Postharvest Biol. Tec. 2020, 159, 111038. https://doi.org/10.1016/j.postharvbio.2019.111038
- Yi, Y.; Luan, P.; Liu, S.; Shan, Y.; Hou, Z.; Zhao, S.; Jia, S.; Li, R. Efficacy of Bacillus subtilis XZ18-3 as a biocontrol agent against Rhizoctonia cerealis on wheat. Agriculture 2022, 12, 258. https://doi.org/10.3390/agriculture12020258
- Diao, W.; Hu, Q.; Zhang, H.; Xu, J. Chemical composition, antibacterial activity and mechanism of action of essential oil from seeds of fennel (Foeniculum vulgare Mill.). Food Control 2014, 35, 109-116. https://doi.org/10.1016/j.foodcont.2013.06.0561221
- Question: The in vivo efficacy was screened using only a few conditions. Only curative efficacy was assayed? And preventive efficacy? What was the concentration of SFB (CFU/mL) in the sprayed solution?
Response: We are grateful for the suggestion. So far, we have only carried out the experiment of curative efficacy. In Section 2.8, we have stated that the concentrations of sterile supernatant (SS) used in the in vivo control effect experiment were 100 mL/L, 200 mL/L and 400 mL/L, respectively. The concentration here refers to the concentration of SS diluted with sterile water, therefore the unit is mL/L.
- Question: The discussion must be extended by compering this study with other reported methods.
Response: We are grateful for the suggestion. In the modified manuscript, we have rewritten the “Discussion” section and added references to fully discuss the results in our study (L342-373).
As mentioned above, we tried our best to improve the manuscript and made some changes in the revised manuscript. These changes will not influence the content and framework of the manuscript.
We appreciate for Mr. Randal Zhang, reviewers and editors' warm work earnestly, and hope that the correction will meet with approval.
Once again, thank you very much for your comments and suggestions.

Round 2
Reviewer 2 Report
Nice work. I recommend accept the manuscript.
Reviewer 4 Report
It's great that You added figure 7. The discussion has been significantly enriched and shows the contribution of other researchers to the topic.